∂ | **Open Peer Review** | Virology | Research Article

# Influenza A virus infection alters the resistance profile of gut microbiota to clinically relevant antibiotics

Marina Robas,[1] Jesús Presa,[1] Javier Arranz-Herrero,[1,2] Soner Yildiz,[3,4] Sergio Rius-Rocabert,[1,2,5] Francisco Llinares-Pinel,[1] Agustin Probanza,[1] Mirco Schmolke,[3,6] Pedro A. Jiménez,[1] Estanislao Nistal-Villan[1,5]

**ABSTRACT** Influenza A virus (IAV) infection triggers quantitative and qualitative modifications in lung and intestinal microbiota composition, which contain an important reservoir of antibiotic resistance genes. Analysis of genetic changes is a common practice in studies that analyze microbiota modifications. However, there is little evidence of functional changes linked to such microbiota modifications. This study evaluates some cecal microbiota's functional changes, comparing sublethal IAV with mock-infected mice. Community-wide phenotypic metabolic profile (Biolog EcoPlatesTM) and relative antibiotic resistance changes to clinically relevant antibiotics (cenoantibiogram) have been performed in this context. Results reveal a temporal association between IAV infection and alterations in nutrient substrate profile usage as well as changes in antibiotic resistance of cecal microbiota. Alterations are transient and predominantly occur at early time points post-IAV infection. There is a functional rebalance in nutrient substrate usage and antibiotic resistance under the established culture conditions, accompanied by a decrease in microbial density of the cecal community on days 5 and 7 after the IAV infection. Our data underline that active IAV infections altering microbial populations are associated with changes in nutrient usage preferences and affect community behaviors toward specific antibiotics. These findings could have implications including activation of nutrient-related metabolic stress at the microbiota community level and additional antibiotic resistance selection mechanisms of clinically relevant infections.

**IMPORTANCE** Influenza virus infection affects both lung and intestinal bacterial community composition. Most of the published analyses focus on the characterization of the microbiota composition changes. Here we assess functional alterations of gut microbiota such as nutrient and antibiotic resistance changes during an acute respiratory tract infection. Upon influenza A virus (IAV) infection, cecal microbiota drops accompanied by a decrease in the ability to metabolize some common nutrients under aerobic conditions. At the same time, the cecal community presents an increase in resistance against clinically relevant antibiotics, particularly cephalosporins. Functional characterization of complex communities presents an additional and necessary element of analysis that nowadays is mainly limited to taxonomic description. The consequences of these functional alterations could affect treatment strategies, especially in multimicrobial infections.

**KEYWORDS** intestinal microbiota, influenza virus, cenoantibiogram, microbial community, antibiotic resistance

Address correspondence to Pedro A. Jiménez, pedro.jimenezgomez@ceu.es, or Estanislao Nistal-Villan, estanislao.nistalvillan@ceu.es.

Marina Robas and Jesús Presa contributed equally to this article. The order is based on the number of hours dedicated to each task.

The authors declare no conflict of interest.

See the funding table on p. 16.

The human intestinal microbiota plays important roles in the body (1, 2) including maintaining the integrity of the intestinal mucosa, avoiding colonization of opportunistic pathogens, and developing and maintaining the immune system (3), or

synthesis of vitamins and metabolites, like anti-inflammatory molecules such as butyrate (4). Alterations of the intestinal microbiota are associated with a higher incidence of gastrointestinal disorders, cancer, cardiovascular, neurological, respiratory (2), and metabolic diseases such as obesity or type 2 diabetes (4), among others.

The gut microbiota composition is relatively stable during adulthood (2, 4), although between 30% and 40% of the composition can be modified through either the influence of hormonal, genetic, and environmental factors such as diet (1, 2, 4), geographic location, exposure to toxins and carcinogens (2), infectious diseases (5–7) antibiotic consumption, or the immune system (8), leading to a state of dysbiosis (2). However, less is known about the consequences of acute viral infections on microbiota composition and functionality.

Influenza viruses are associated with hundreds of thousands of deaths worldwide every year (9). Gut dysbiosis during influenza virus infection may influence the pathogenesis associated with secondary bacterial pneumonia (10). Treatment of these complications can be addressed with a combination of antibiotics and respiratory support therapy.

Traditional analysis of bacteria properties focuses on the metabolic and functional features at the species or strain levels. One example of a functional analysis is the antibiogram. However, functional analysis of complex communities, such as the intestinal microbiota, adds relevant information that cannot be obtained by adding properties of each microorganism and even less by the exact taxonomic composition description (11). Influenza A virus (IAV) infections trigger changes in the respiratory and gut microbiota (10, 12–15). Most of the time, bacterial community analysis is restricted to the identification of microorganisms present in the samples. Importantly, gut commensal bacteria adapt rapidly to changing environmental conditions, mostly by regulating their gene expression profile (16). Functional properties of the gut microbiota during or after IAV infection, beyond metagenomic analysis in bacteria composition, have not been well explored.

One approach to determining the metabolic microbial profiles of communities associated with environmental changes is the analysis of their community-level physiological profile (CLPP) using Biolog EcoPlates (11, 17). Analysis of the antibiotic resistance of bacterial communities is an additional functional characterization assay enabling the study of changes in the equilibrium of complex bacterial communities. Characterization of antibiotic resistance is especially relevant in the context of Influenza A virus infection since it favors secondary or superinfections with bacterial pathogens. From a global perspective, antibiotic resistance is considered one of the biggest public health challenges (18, 19).

Antibiotic resistance is not a static property. Resistance is developed as an adaptive response of bacteria to different environmental conditions (20). Determination of mechanisms driving the evolution of antibiotic resistance is important in establishing novel strategies to counteract their effects (21, 22). Few studies have focused on characterizing antibiotic resistance profiles in microbial ecological communities (23, 24). In this context, the cenoantibiogram assay has been developed to determine the community-wide phenotypic resistance profile to clinically relevant antibiotics (17).

In this work, we applied a combination of metabolic and antibiotic resistance profiling to cecal microbial communities of mice exposed to IAV or mock infection. We used samples from IAV-infected animals previously described by Yildiz et al. (15). Under aerobic manipulation conditions, CLPP analysis of samples indicates changes in overall carbon and nitrogen nutrient preferences. Cenoantibiogram analysis of the different groups of samples indicates additional alterations in antibiotic resistance of the cecal bacteria communities to specific antibiotics. These results indicate not only the influence of the viral infection on changes in the intestinal microbiota functionality but also suggest underlying mechanisms that could drive the selection of antibiotic-resistant bacteria.

## RESULTS

### Semiquantitative determination of bacteria in cecal samples

Previous reports have described composition changes in gut microbiota after IAV infection. However, it remains unclear whether those changes are associated with relevant phenotypic differences. In the present work, cecal samples from mock or IAV-infected mice described by Yildiz et al. (15) were used to study functional changes associated with the infection.

Comparison of the relative changes and content of viable bacteria in mock- or IAV-infected mice was performed on different days post-infection (3, 5, 7, and 14 dpi) by the extraction of bacterial content from different samples in a fixed amount of cecal material that was resuspended in saline solution buffer. Bacteria in suspension were cultured in a cap-closed laminoculture tube to estimate the relative number of colony-forming bacteria per gram of cecal material (Fig. 1). Under this experimental setting, the relative number of microorganisms in mock control groups was estimated as $10^6$ CFU/g. A significant decrease in cecal bacterial growing load was observed on day 3 ($10^5$ CFU/g), day 5, and day 7 ($10^4$ CFU/g) in IAV-infected mice. On day 14, there was a recovery tendency in the number of colony-forming bacteria in the cecal samples of IAV-infected animals. This decrease in the number of colonies growing in cap-closed laminoculture tubes correlates with a drop in genome counts in the small intestine samples previously reported (15) from the same animals. This drop also coincides with high viral titers in the lungs of IAV-infected animals at days 3 and 7 post-inoculation as described in Yildiz et al (15).

### Analysis of CLPPs in microbial communities

Alterations of bacterial communities in the intestinal microbiota during viral infection have been associated with reduced food intake as typical sickness behavior (10, 14). Alteration of a bacterial community composition may imply the need to switch metabolic states, to use different nutrients. To test this hypothesis, Biolog EcoPlates were used to determine the CLPP by analyzing the consumption of a set of defined substrates within cecum material sampled from IAV and mock-infected animals (see Table S1 at https://figshare.com/articles/dataset/Supplementary_Table_1_biolog_Spectrum_xlsx/24428692). After standardizing the number of inoculated bacteria, Biolog EcoPlates

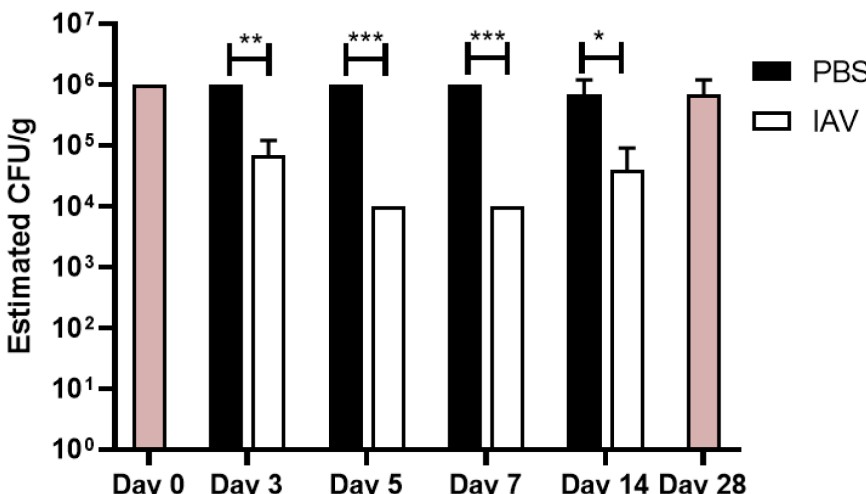

**FIG 1** Estimation of bacterial density in different experimental groups. Estimation of viable bacterial concentration in 0.4 g of cecal samples resuspended in 40 mL of saline solution. An approximation of bacterial concentration was estimated by observing colony density in CLED from the Uricult laminoculture test. Baseline D0 and Baseline D28 indicate untreated mice sacrificed on D0 or D28, respectively (*n* = 3). Circles represent samples coming from mock-infected mice and squares the IAV-infected mice.

plates were inoculated under aerobiotic conditions and incubated at 37°C for 72 h, the estimated time necessary by these microbial communities to reach the stationary phase of growth. Colorimetry in these plates allows the determination of microbial growth by measuring the absorbance at 590 nm in the presence of individual substrates.

Differences in the consumption of each substrate were analyzed by a random forest methodology using the colorimetric assay values obtained for each mouse sample. To find the substrates that can be differentially metabolized, Boruta analysis was performed (Fig. 2A). Certain substrates, such as α-D-Lactose, D-Xylose, Tween 80, and D-Cellobiose, present statistically relevant differences (green and yellow whiskers) while the rest of the substrates (red whisker plots) did not.

The random forest approach followed by Boruta's analysis of the data does not reveal changes between IAV- and mock-infected animals at particular time points. To find them, a daily comparison was performed by calculating the difference between the average of the substrate consumption indicator for IAV- and the mock-infected groups for each substrate (Fig. 2B). Bacteria density-related changes observed using cap-closed laminocultures occur in infected animals between days 3 and 7 (coincidental with changes observed in animals by Yildiz et al. (15), particularly the high viral titers at days 3 and 7). Therefore, the analysis of substrate use was focused on cecal samples coming from days 3, 5, and 7.

The difference between the average in IAV- and mock-infected groups indicates a drop in consumption of some substrates by the cecal microbiota mainly on day 3, and recovering on days 5 and 7 under our experimental conditions. Together with a drop in consumption of D-Lactose, samples from IAV-infected mice show a drop in consumption of both, simple sugars (i.e., Glucose-1-phosphate, N-acetyl-D-glucosamine, or methyl glucoside) and complex sugars (i.e., xylose, cyclodextrin, or glycogen), and carbon sources like mannitol and the Krebs cycle precursor pyruvate. A small reduction in amino acid arginine consumption was also observed in samples from IAV-infected mice. A potential decrease in the consumption of lipids in the IAV group on day 3 can also be suggested (i.e., a modest drop for Tween 40). At day 5, substrates that can feed the Krebs cycle such as alpha-cyclodextrin, erythritol, methyl-pyruvic-ester, N-acetyl-D-glucosamine, or lactose present a lower consumption average in IAV-infected as compared to mock-infected mice. Interestingly, there is an increase in the utilization of some substrates such as alpha-cyclodextrin, glycogen, or xylose preferentially at day 7 post-infection in IAV-infected mice as compared to the mock-infected ones.

In Table S1 at https://figshare.com/articles/dataset/Supplementary_Table_1_biolog_Spectrum_xlsx/24428692, we can observe a strong dispersion of values for each substrate utilization. A non-parametric Wilcoxon-Mann-Whitney statistical analysis test was used in this case to determine differences in nutrient substrate uptake between IAV- and mock-infected groups at each time point (Fig. 2C). This analysis presents statistically significant IAV- vs mock-infected groups at 3 dpi for D-lactose, methyl-pyruvic ester, or putresin. Significant differences between groups 5 dpi appear for D-lactose and D-galacturonic acid, and 7 dpi for alpha-cyclodextrin and D-xylose. These data suggest a broad metabolic adaptation of cecal microbiota to the environmental changes during acute viral infection of the respiratory tract.

## Cenoantibiogram analysis

Characterization of changes in antibiotic resistance of multi-species communities is a challenge in the field due to several variables that are difficult to address experimentally. For instance, antibiotic resistance phenotypes can be modulated by several factors, including additive and synergistic effects between strains, quorum-sensing/quorum-quenching processes affecting gene expression, as well as the response to biotic and abiotic factors coming from the host. Moreover, antibiotic resistance cannot be explained just by the presence of individual bacterial populations and cannot be inferred from metagenomic analysis approaches like 16S gene sequencing. On the other hand, previous reports have shown that metabolic shifts are one trigger for the development

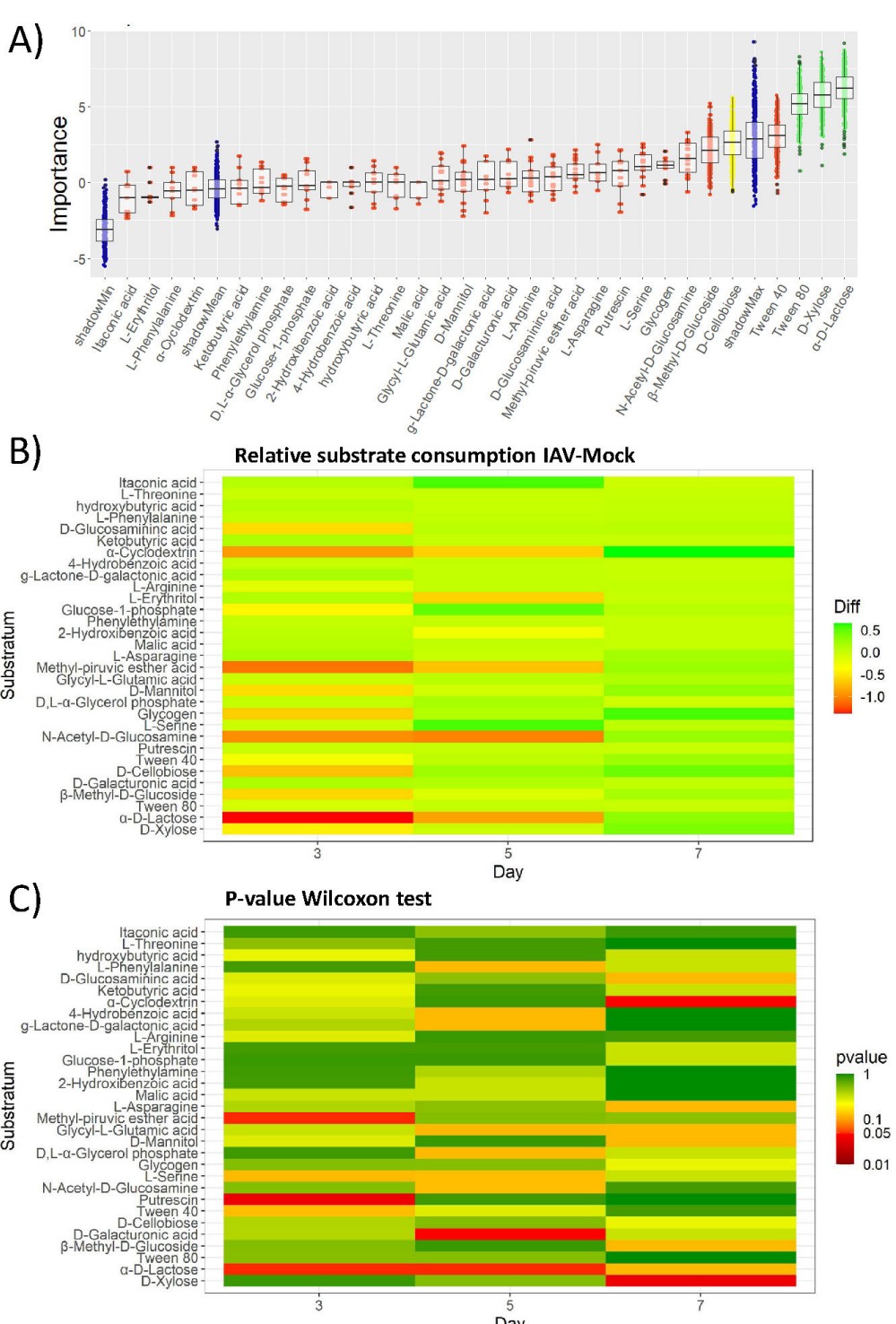

**FIG 2** Changes in the metabolic profile of different cecal microbiota samples extracted from IAV- and mock-infected mice. Nutrient usage profiles are determined using Biolog EcoPlates. (A) Evaluation of significant differences in nutrient usage between the samples under study analyzing the data by an initial random forest iterative approach followed by a Boruta statistical analysis comparing all the measurements from individual nutrients. The importance of each nutrient is represented in a whisker-box plot. Boxes contain 50% of all aleatory arrangements close to the mean of the importance for each nutrient. The whiskers present the remaining arrangements. Red boxes indicate those nutrients with a null effect. Yellow boxes indicate evidence of changes, while green boxes mark the highest statistically significant differences in the ability to metabolize the indicated nutrients. The blue boxes mark the lower, mean, and shadowmax values in the Boruta analysis which also sets the

**FIG 2** (Continued)

cutoff limit. Green boxes correspond to statistically differential values with a $P \leq 0.05$. Each condition contains the analysis of fecal samples from three mice. (B) Differences between the mean of IAV- and the mock-infected ones for each substrate usage in Biolog analysis. Metabolization of each substrate in the Biolog is performed using individual cecal ($n = 3$). Individual values are presented in Table S1 at https://figshare.com/articles/dataset/Supplementary_Table_1_biolog_Spectrum_xlsx/24428692 (Biolog). Results are represented in AWCD (average well-color development) where the color gradient reference represents arbitrary difference units. Positions close to red represent the largest drops in nutrient usage when comparing IAV- vs mock-infected mice and in green the largest increases. (C) Representation of the $P$-value corresponding to the analysis of the differences between the IAV- and mock-infected groups for each of the substrates. The analysis was performed using the non-parametric Wilcoxon-Mann-Whitney analysis test. Results compare samples collected on days 3, 5, and 7. M: mock; I: IAV infections.

of bacterial antibiotic resistance (25). Hence, how changes in microbiota composition or phenotype could affect the overall antibiotic resistance of bacterial communities has not been well understood.

The entire community resistance profile against a particular antibiotic within the host microbiota will be shaped by the cumulative effect of different resistance mechanisms expressed by cohabitating microorganisms composing a community at a particular time within a dynamic environment. We designate the term *cenoantibiogram* to express a specific community's antibiotic profile.

Cenoantibiogram profiles were determined as described in the Section Materials and Methods, by diluting the bacterial suspension samples 1:3 in saline solution, which was inoculated on Vitek 2 AST-N243 cards and automatically analyzed. In Table 1, we can observe the minimal inhibitory concentration (MIC) of different antibiotics for the bacteria present within cecal samples from IAV-infected and mock groups under the given experimental conditions. To interpret changes in antibiotic resistance, *Escherichia coli* was set as the reference bacteria for the analysis comparison of all the samples. Accordingly, MIC values of this reference allow evaluation of changes in antibiotic resistance between samples, that is, between groups or the same group over time.

Table 1 presents the antibiotic resistance profile changes across time points between IAV- and mock-infected groups. Specifically, the pattern of increased ESBL activity appears mainly positive in cecal samples from IAV-infected mice starting on day 3 and is generalized in all IAV-infected mice, but not in their mock controls, except for one animal, on days 5 and 7. On day 14, the cecal community behavior tends to recover resistance pattern previous to infection. Interestingly, on day 7 post-infection, only two out of six samples from IAV-infected mice presented an increased relative resistance against the β-lactam amoxicillin as compared to the mock samples. In any case, the amoxicillin resistance in the microbiota of these two cecal samples disappeared in the presence of the β-lactamase inhibitor clavulanic acid.

Another interesting phenotype observed on day 3 post-infection was the increased relative antibiotic resistance against some second-generation cephalosporins such as cefoxitin (6/6) in samples from IAV-infected mice. These differences were diminished at later time points. A similar effect was observed for resistance against third- (ceftazidime) (5/6) and fourth-generation (cefepime) cephalosporins (4/6). An increase in resistance against the carbapenem imipenem was also observed in three over six animals on day 3 post-IAV infection. Increased relative resistance against aminoglycosides amikacin (4/6) and gentamicin (3/6) can be observed in samples from some animals at day 3; however, this phenotype seems to disappear at later time points. Interestingly, fluctuations in the antibiotic resistance against quinolones, particularly in resistance against nalidixic acid increased in the IAV group (5/6 mice) compared to mock on day 3 post-infection. These changes, however, were contradictory when compared to the relative resistance in the samples from the basal group on day 0. In any case, differences between IAV- and mock-infected groups disappeared at later time points suggesting a transient impact of IAV infection on the global resistance profile of gut microbiota. There appeared to be little alteration in antibiotic resistance against tetracyclines such as tigecycline,

**TABLE 1** MIC values (µg/mL) of intestinal microbiota samples under study using *E. coli* as reference bacteria[a]

| Time after infection (days) | Group | ESBL | Penicillins | | | Cephalosporins | | | | | | Carbapenem | Aminoglycosides | | Quinolones | | Tetracyclines |
|---|---|---|---|---|---|---|---|---|---|---|---|---|---|---|---|---|---|
| | | | Ampicillin | Amoxicilin/ clavulanic Ac. | Piperacillin/ tazobactam | 2nd Cefuroxime axetil | 2nd Cefuroxime | 2nd Cefoxitin | 3rd Cefotaxime | 3rd Ceftazidime | 4th Cefepime | Imipenem | Amikacin | Gentamicin | Nalidixic Ac. | Ciprofloxacin | Tigecycline |
| 0 | B | Neg | ≤2 | ≤2 | ≤1 | 2 | 2 | 16 | 4 | 4 | 2 | ≤0.25 | ≤2 | ≤1 | ≥32 | 0.5 | ≤0.5 |
| | B | Neg | 4 | ≤2 | ≤1 | 4 | ≤1 | ≤4 | 2 | 8 | 8 | 0.5 | 16 | ≤1 | ≥32 | 0.5 | ≤0.5 |
| | B | Neg | ≤2 | ≤2 | ≤1 | ≤1 | ≤1 | ≥64 | 2 | 8 | 8 | 0.5 | 8 | ≤1 | ≥32 | 0.5 | ≤0.5 |
| | B | Neg | ≤2 | ≤2 | ≤1 | 4 | 2 | ≥64 | 2 | 8 | 2 | ≤0.25 | 8 | ≤1 | ≥32 | 0.5 | ≤0.5 |
| 3 | M | Neg | ≤2 | ≤2 | ≤1 | 2 | 2 | ≤4 | 4 | 8 | 2 | ≤0.25 | 4 | ≤1 | 15 | 0.5 | ≤0.5 |
| | M | Neg | ≤2 | ≤2 | ≤1 | 2 | 2 | ≤4 | 4 | 8 | 2 | ≤0.25 | 4 | ≤1 | 15 | 0.5 | ≤0.5 |
| | M | Neg | ≤2 | ≤2 | ≤1 | 2 | 2 | ≤4 | 4 | 8 | 2 | ≤0.25 | 4 | ≤1 | 15 | 0.5 | ≤0.5 |
| | M | Pos | ≤2 | ≤2 | ≤1 | ≤1 | ≤1 | ≤4 | 2 | 8 | ≤1 | ≤0.25 | ≤2 | ≤1 | 16 | 0.5 | ≤0.5 |
| | M | Neg | ≤2 | ≤2 | ≤1 | ≤1 | ≤1 | ≤4 | 2 | 8 | ≤1 | ≤0.25 | ≤2 | ≤1 | 16 | 0.5 | ≤0.5 |
| | M | Neg | ≤2 | ≤2 | ≤1 | ≤1 | ≤1 | ≤4 | 2 | 8 | ≤1 | ≤0.25 | ≤2 | ≤1 | 16 | 0.5 | ≤0.5 |
| | IAV | Neg | ≤2 | ≤2 | ≤1 | 4 | 4 | ≥64 | ≤1 | 16 | 8 | 1 | 16 | ≤1 | ≥32 | 0.5 | ≤0.5 |
| | IAV | Pos | ≤2 | ≤2 | ≤1 | ≤1 | ≤1 | ≥64 | ≤1 | 16 | 2 | 0.5 | ≤2 | ≤1 | ≥32 | 0.5 | ≤0.5 |
| | IAV | Neg | ≤2 | ≤2 | ≤1 | ≤1 | ≤1 | ≥64 | 8 | 8 | 8 | 0.5 | 4 | ≤1 | 16 | 0.5 | ≤0.5 |
| | IAV | Pos | ≤2 | ≤2 | ≥4 | 4 | 4 | ≥64 | ≤1 | 16 | 4 | ≤0.25 | 16 | 4 | ≥32 | 0.5 | ≤0.5 |
| | IAV | Pos | ≤2 | ≤2 | ≥4 | 4 | 4 | 32 | ≤1 | 16 | 2 | ≤0.25 | 32 | 4 | ≥32 | 0.5 | ≤0.5 |
| | IAV | Pos | ≤2 | ≤2 | ≥4 | 4 | 4 | ≥64 | ≤1 | 16 | 8 | ≤0.25 | 32 | 4 | ≥32 | 0.5 | ≤0.5 |
| 5 | M | Neg | ≤2 | ≤2 | ≤1 | ≤1 | ≤1 | ≤4 | ≤1 | ≤1 | 1 | ≤0.25 | 8 | 2 | ≥32 | 0.5 | ≤0.5 |
| | M | Neg | ≤2 | ≤2 | ≤1 | ≤1 | ≤1 | ≤4 | ≤1 | ≤1 | 1 | ≤0.25 | 8 | ≤1 | ≥32 | 0.5 | ≤0.5 |
| | M | Pos | ≤2 | ≤2 | ≤1 | ≤1 | ≤1 | ≤4 | ≤1 | ≤1 | 1 | ≤0.25 | 8 | ≤1 | ≥32 | 0.5 | ≤0.5 |
| | M | Neg | ≤2 | ≤2 | ≤1 | ≤1 | ≤1 | ≤4 | ≤1 | ≤1 | 1 | ≤0.25 | 8 | ≤1 | ≥32 | 0.5 | ≤0.5 |
| | M | Neg | ≤2 | ≤2 | ≤1 | ≤1 | ≤1 | ≤4 | ≤1 | ≤1 | 1 | ≤0.25 | 8 | ≤1 | ≥32 | 0.5 | ≤0.5 |
| | M | Neg | ≤2 | ≤2 | ≤1 | ≤1 | ≤1 | ≤4 | ≤1 | ≤1 | 1 | ≤0.25 | 8 | ≤1 | ≥32 | 0.5 | ≤0.5 |
| | IAV | Pos | ≤2 | ≤2 | ≤1 | ≤1 | ≤1 | ≥64 | 4 | 4 | ≤1 | ≤0.25 | ≤2 | ≤1 | ≥32 | 0.5 | ≤0.5 |
| | IAV | Pos | ≤2 | ≤2 | ≤1 | ≤1 | ≤1 | 8 | ≤1 | 8 | ≤1 | ≤0.25 | ≤2 | 1 | ≥32 | 1 | 4 |
| | IAV | Pos | ≤2 | ≤2 | ≤1 | ≤1 | ≤1 | <=4 | ≤1 | 8 | ≤1 | ≤0.25 | ≤2 | ≤1 | 4 | 0.5 | ≤0.5 |
| 7 | M | Neg | ≤2 | ≤2 | ≤1 | 4 | 4 | ≥64 | 2 | 8 | 8 | ≤0.25 | ≤2 | ≤1 | ≥32 | 0.5 | ≤0.5 |
| | M | Neg | ≤2 | ≤2 | ≤1 | ≤1 | ≤1 | 8 | ≤1 | 8 | 8 | 1 | 8 | ≤1 | ≥32 | 0.5 | ≤0.5 |
| | M | Neg | ≤2 | ≤2 | ≤1 | ≤1 | ≤1 | 8 | ≤1 | 16 | 8 | 1 | 16 | ≤1 | ≥32 | 2 | ≤0.5 |
| | IAV | Pos | 16 | ≤2 | ≤1 | 32 | 32 | ≥64 | ≤1 | 8 | 8 | 0.5 | 8 | ≤1 | ≥32 | 0.5 | ≤0.5 |
| | IAV | Pos | ≤2 | ≤2 | ≤1 | 4 | 4 | ≤4 | 2 | 4 | ≤1 | ≤0.25 | ≤2 | ≤1 | ≥32 | 0.5 | ≤0.5 |
| | IAV | Pos | 16 | ≤2 | ≤1 | ≤1 | ≤1 | 16 | 4 | 8 | ≤1 | ≤0.25 | 4 | 2 | ≥32 | ≥4 | 1 |
| | IAV | Pos | ≤2 | ≤2 | ≤1 | 4 | 4 | ≤4 | ≤1 | ≤1 | ≤1 | ≤0.25 | ≤2 | ≤1 | ≤2 | ≤0.25 | ≤0.5 |
| | IAV | Pos | ≤2 | ≤2 | ≤1 | ≤1 | ≤1 | ≤4 | ≤1 | 2 | ≤1 | ≤0.25 | ≤2 | ≤1 | ≤2 | ≤0.25 | ≤0.5 |

**TABLE 1** MIC values (µg/mL) of intestinal microbiota samples under study using *E. coli* as reference bacteria[a] (*Continued*)

| Time | Group | ESBL | Penicillins | | | | | Cephalosporins | | | | Carbapenem | Aminoglycosides | | | Quinolones | Tetracyclines |
|---|---|---|---|---|---|---|---|---|---|---|---|---|---|---|---|---|---|
| 14 | IAV | Pos | ≤2 | ≤2 | ≤1 | 4 | 4 | ≤1 | 4 | ≤1 | ≤4 | ≤0.25 | ≤2 | 4 | ≤1 | ≤0.25 | ≤0.5 |
| | M | Pos | ≤2 | ≤2 | ≥4 | ≤1 | ≤1 | ≤1 | 2 | ≤1 | ≤4 | ≤0.25 | ≤2 | 8 | ≤1 | ≤0.25 | ≤0.5 |
| | M | Neg | ≤2 | ≤2 | ≥4 | ≤1 | ≤1 | 4 | 8 | 2 | ≤4 | ≤0.25 | 4 | ≥32 | ≤1 | ≤0.25 | ≤0.5 |
| | M | Pos | 4 | 4 | ≥4 | ≤1 | ≤1 | ≤1 | 2 | 2 | ≤4 | ≤0.25 | 4 | ≥32 | ≤1 | 1 | ≤0.5 |
| | IAV | Neg | ≤2 | ≤2 | ≤1 | 2 | 2 | 2 | 4 | ≤1 | 8 | ≤0.25 | 8 | ≥32 | 2 | 0.5 | ≤0.5 |
| | IAV | Pos | ≤2 | ≤2 | ≤1 | 4 | 4 | 2 | 16 | 4 | ≥64 | 0.5 | 16 | ≥32 | ≤1 | ≥4 | 1 |
| | IAV | Neg | ≤2 | ≤2 | ≤1 | 2 | 2 | 2 | 4 | ≤1 | 8 | ≤0.25 | 8 | ≥32 | 2 | 0.5 | ≤0.5 |
| | IAV | Pos | ≤2 | ≤2 | ≥4 | 2 | 2 | 8 | 16 | 8 | ≥64 | ≤0.25 | 8 | ≥32 | 2 | 2 | 4 |
| | IAV | Neg | ≤2 | ≤2 | ≥4 | 2 | 2 | 2 | 8 | ≤1 | ≥64 | ≤0.25 | ≤2 | ≥32 | ≤1 | 0.5 | ≤0.5 |
| | IAV | Neg | ≤2 | ≤2 | ≥4 | ≤1 | ≤1 | 2 | 8 | ≤1 | ≥64 | ≤0.25 | ≤2 | ≥32 | ≤1 | 0.5 | ≤0.5 |
| 28 | B | Neg | ≤2 | ≤2 | ≤1 | 4 | 4 | 8 | 16 | ≤1 | ≥64 | 0.5 | 8 | ≥32 | ≤1 | 0.5 | ≤0.5 |
| | B | Neg | ≤2 | ≤2 | ≤1 | 8 | 8 | 8 | 16 | 2 | ≥64 | ≤0.25 | ≤2 | ≥32 | ≤1 | 1 | ≤0.5 |
| | B | Pos | ≤2 | ≤2 | ≤1 | 4 | 4 | 8 | 4 | ≤1 | 32 | ≤0.25 | ≤2 | ≥32 | ≤1 | 1 | ≤0.5 |
| | B | Neg | 4 | 4 | ≤1 | ≤1 | ≤1 | 4 | 16 | 2 | ≥64 | ≤0.25 | 8 | ≥32 | ≤1 | ≥4 | ≤0.5 |
| | B | Neg | ≤2 | ≤2 | ≥4 | ≤1 | ≤1 | ≤1 | ≤1 | ≤1 | 32 | ≤0.25 | 4 | 16 | ≤1 | 0.5 | ≤0.5 |
| | B | Neg | 4 | 4 | ≤1 | ≤1 | ≤1 | 2 | 4 | ≤1 | ≤4 | ≤0.25 | 4 | ≥32 | ≤1 | 1 | ≤0.5 |

[a]Mock- or IAV-inoculated mice were sacrificed at the indicated times. Cecal samples were harvested and quickly frozen. Samples were analyzed by dissolving 0.4 g of cecal material in 40 mL saline solution. After homogenizing, samples were clarified by centrifugation to separate bacteria from macroscopic material. 1 mL of bacteria suspension was mixed with 2 mL of saline solution to prepare the sample suspension for inoculation into the Vitek 2 AST-N243 cards. MIC was determined by interpreting the Vitek 2 card results using *Escherichia coli* as a standard reference. "IAV," mice inoculated with Influenza A virus. ESBL: extended-spectrum β-lactamases. Pos: compatible with a resistance profile for an ESBL. Neg: not compatible with a resistance profile for an ESBL.

some relative resistance against this antibiotic increased in two animals on day 14 after infection.

Statistical analysis of the results presents a challenge due to the low number of samples and strong dispersion of the data. To determine changes in the antibiotic resistance profile in any sample at any point, antibiotic data were also analyzed by a random forest approach followed by Boruta analysis (Fig. 3A). Significant antibiotic resistance alterations were observed for most of the antibiotics tested (green boxes). The largest differences between groups can be attributed to the presence of extended-spectrum beta-lactamases (ESBL) phenotype, including changes in resistance against ampicillin, piperacillin/tazobactam, or cephalosporins. Additional changes in resistance were observed also in the case of quinolones such as nalidixic acid or ciprofloxacin, and aminoglycosides such as amikacin.

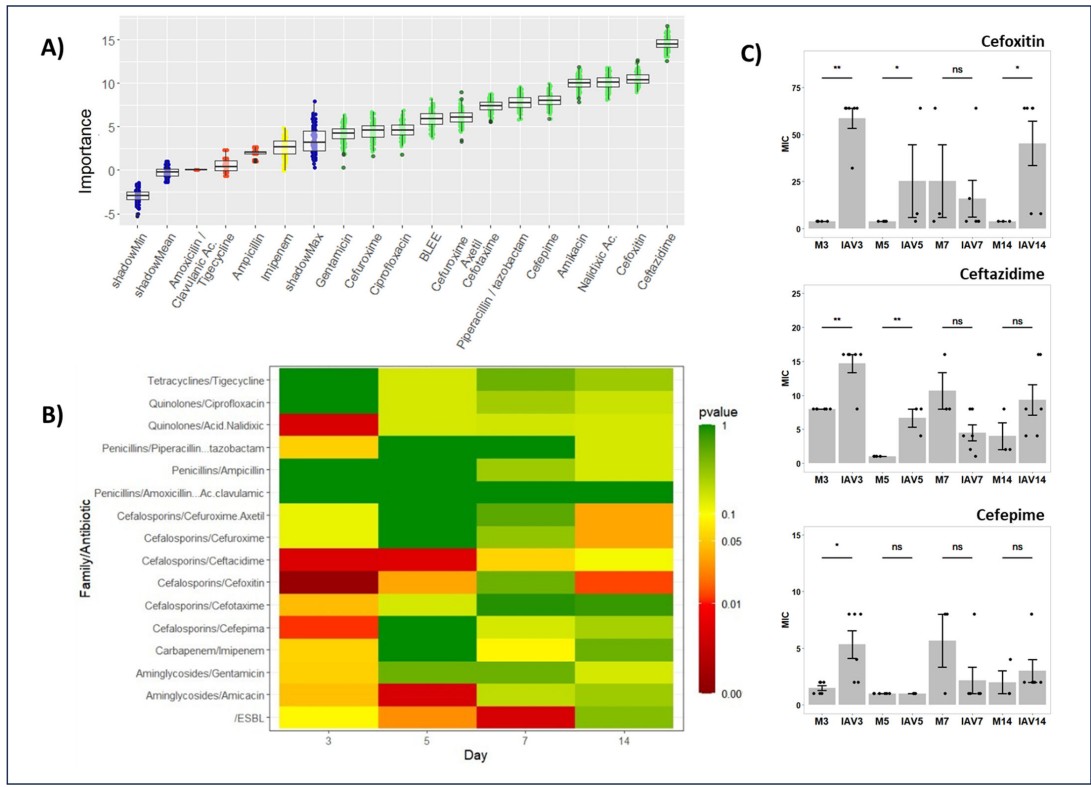

**FIG 3** Antibiotic resistance changes in cecal microbiota. The cenoantibiogram approximation of antibiotic resistance profiles was determined with a Vitec 2 automatic analyzer using AST-N243 cards. (A) Detection of significant differences in the samples under study was performed using a random forest iterative approach followed by a Boruta statistical analysis comparison. The importance of each change in resistance to different antibiotics is represented in a whisker-box plot. Boxes contain 50% of all aleatory arrangements close to the mean of the importance for each antibiotic resistance change. The whiskers present the remaining arrangements. Red boxes indicate those nutrients with a null effect. Yellow boxes indicate evidence of changes, while green boxes mark the highest statistically significant differences in the ability to metabolize indicated nutrients. The blue boxes mark the lower, mean, and shadowmax values in the Boruta analysis which also sets the cutoff limit ($P \leq 0.05$). The whiskers in the panel represent the attribute importance over the different random forest arrangements followed after the Boruta analysis. Each condition contains the analysis of fecal samples from 3 to 6 mice. ESBL: Extended-spectrum beta lactamases. (B) Data represent a $P$-value differences heat map in antibiotic resistance changes in cecal samples from the IAV- and mock-infected mice groups using the non-parametric Wilcoxon-Mann-Whitney analysis test. Results compare samples collected on days 3, 5, 7, and 14 post-infection. Yellow to brown colors represent situations where the antibiotic resistance against the indicated antibiotic significantly increases in samples from IAV-infected mice at each particular time point. Blue and purple colors $P$-value > 0.05 (without statistical differences); green, yellow, orange, and red colors $P$-value $\leq 0.05$ (significant differences). (C) Representation of specific cephalosporin antibiotic resistance variation comparing the average and standard deviation of the minimal inhibitory concentration (MIC) using *E. coli* as reference bacteria against cephalosporins cefoxitin, ceftazidime, and cefepime. Individual values are presented in Table 1. Statistical analysis performed using the non-parametric Wilcoxon-Mann-Whitney test (\*: $P < 0.05$, \*\*: $P < 0.01$). The number of mice for each condition ranges from $n = 3$ to $n = 6$.

To assess statistical differences in microbiota communities for individual sampling days, an additional Wilcoxon-Mann-Whitney statistical analysis was performed by comparing antibiotic resistance changes of fecal samples between IAV- and mock-infected mice groups at different days post-infection. Figure 3B shows the *P*-values from this analysis. This examination reveals no differences or minor, not statistically relevant differences in most of the comparisons. Comparisons represented with orange and red colors with *P*-values ≤ 0.05 were considered significantly different. Changes in the relative antibiotic resistance increase were more pronounced in IAV-infected animals as compared to mock-infected animals early in infection (3 dpi). At this time point, changes in relative antibiotic resistance significantly changed for ESBL phenotype as well as resistance against antibiotics such as gentamycin, amikacin, ceftazidime, cefoxitin, or cefepime. Differences were not always sustained over time, yet specifically, ESBL phenotype relative resistance in infected animals was further elevated on day 5 and even more strongly on day 7 post-IAV infection. A significant increase in relative antibiotic resistance to certain cephalosporins was observed in fecal samples from IAV-infected mice as compared to the mock samples. This was observed for some second-, third-, and fourth-generation cephalosporins (Fig. 3C) on day 3, as well as on day 5 post-infection in the case of second-generation cefoxitin and third-generation ceftazidime cephalosporins. Interestingly, some relative antibiotic resistances present different kinetics. For instance, an initial increasing resistance tendency to second-generation cephalosporins cefuroxime and cefuroxime-acetyl was observed on day 3 and disappeared on days 5 and 7. This resistance increased again on day 14 in the IAV as compared to mock-infected groups. Other antibiotic resistance changes were also observed; however, no biological or statistical relevance was attributed to them.

## DISCUSSION

Acute and chronic viral infections alter the composition of intestinal microbiota, as described for HIV (26), HCV (27), or SARS-CoV-2 (28, 29). Influenza virus infection also alters intestinal microbiota composition, both in mice (10, 12, 14, 15, 30, 31) and in humans (32). The three main intestinal microbiota changes upon viral infection are as follows: (i) a decrease in the *Bacteroidetes* count (13, 15, 31), (ii) an increase in Proteobacteria (10, 12, 30–32), including *Escherichia coli* (31, 32), and (iii) a decrease in the *Firmicutes* count (14, 15), including the genus *Lactobacillus* (10, 14, 31). These changes are mainly observed on day 7 after infection, recovering the composition on day 14 post-infection, specifically in the small intestine samples from animals used for the assays shown here (10, 15). Bartley et al. (30) and Groves et al. (14) described alterations after 4 days post-infection. These differences can be attributed to particularities between virus strains, animal husbandry, or differences in experimental conditions. Upon infection, driving forces for bacteria community alterations could be attributed to microbiota adaptations to reduced uptake of food and water (10, 14) as well as innate and adaptive immune responses (12, 16).

It is important to know how primary viral infections may influence not just the composition but also the functional properties of the gut microbial community (33). The metabolic state of bacterial communities and/or alterations in antibiotic resistance profiles may influence the expected response against oral antibiotic treatments to prevent secondary bacterial infections. Current approaches to describe microbiota composition do not predict the functional performance of complex microbiota communities.

The metabolic profile in the samples analyzed here (Fig. 2) shows a reduction in the consumption pattern of lactose on days 3 and 5 post-infection as well as different sugars (D-Lactose, Glucose-1-phosphate, N-acetyl-D-glucosamine, methyl glucoside, xylose, cyclodextrin, or glycogen) and other possible carbon sources such as pyruvate, mannitol, or Tween 40. Interestingly, amino acid consumption seems not much affected except for an initial drop in the consumption of arginine on day 3 and a tendency to increase the consumption of serine on day 5 (Fig. 2A).

These changes in nutrient uptake could be attributed to different alternatives that will require novel experimental metabolomic approaches and culture conditions for a better description. Changes in metabolic performance by specific microorganisms, microbiota substitution, or simultaneous influence of both are likely to be behind the observed changes. It is also relevant to notice that these changes are coincidental with a body weight loss between days 5 and 9 post-infection, high viral titers at days 3 and 7, and a drop in core body temperature at day 7 in the mice analyzed as reported by Yildiz et al. (15). For example, nutrient uptake changes could be attributed to possible alterations in *Enterobacteriaceae* belonging to Gamma proteobacteria (31, 32), which could be responsible also for the changes observed in the cenoantibiogram results (ESBL phenotype changes). In this sense, gene expression related to the catabolism of simple sugars can be significantly downregulated in *Escherichia coli* present in the intestinal microbiota during an intestinal inflammatory process (34). By contrast, the expression of some genes associated with amino acid metabolism can be stimulated, like genes involved in L-serine metabolism (34). The observed decrease in lactose and sugar consumption by bacterial communities observed in our study may also be explained by the drop in the number of lactose-consuming bacteria such as *Lactobacillus spp* (8) or *Bifidobacterium spp* (10, 32) described on day 7 post-infection (10) and that could be correlated with a drop in Bacilli in fecal samples from IAV-infected mice (Fig. 4). These bacteria are part of the normal commensal microbiota, which may have been affected during the first days after viral infection. In any case, despite the possible role of specific bacteria, complex functional interactions between members of a microbial community cannot be predicted by a simple analysis of the bacteria present in microbiota like those using just metagenomic analysis. The results presented here highlight the need to increase the attention to microbial community behaviors determined by functional changes.

Metagenomic analysis upstream (small intestine) and downstream (fecal) of cecum microbiota (Fig. 4) presents significant changes at day 7 post-infection. The prevalence of the Muribaculaceae family is reduced at 7 dpi (days post-infection). Lactobacillaceae and Spreptococcaceae families within the Lactobacillares order are increased at 7 dpi. At the same time, the Gammaproteobacteria class presents an increase at 7 dpi. In particular, Pseudomonadaceae, Halothiobacillaceae, Enterobacteriaceae, Pasteurellaceae, Vibrionaceae, and Moraxellaceae families present the highest increase, Verrucomicrobiaceae family within the Verrucomicrobiae class experience an increase also at 7 dpi. These profiles present differences between animals and can also be linked to the high variability of our functional assays of cecal microbiota.

Analysis of fecal microbiota at days 7 and 14 post-infection also presents some differences between IAV- and mock-infected animals. In this case, no clear differences could be observed in Bacteroidia, but a reduction in Lactobacillus at 7 dpi could be observed. A small but significant increase in the Gammaproteobacteria and Verrucomicrobiae classes can be observed at 7 dpi (Panel F of Fig. 4). This correlation between SI and fecal microbiota could indicate an intermediate composition at the cecum, whose samples were used for the functional assays. In particular, Enterobacteriaceae increase in IAV-infected animals both in small intestine and feces correlate with the functional phenotype described. Metagenomic analyses, however, cannot guarantee the identification of the members in the communities responsible for the functional changes analyzed.

Reproducing the cecal microbiota in environmental conditions is challenging. We are aware of the culture conditions limitations in our experimental setup, and how these conditions can affect the interpretations of the results, particularly the aerobic manipulation of the samples. The conditions used are however the same for all samples. The same animals inoculated with IAV suffered a gradual decrease in certain small intestine bacteria that peaked on days 5–7 post-infection (15) compared to mock controls. This effect was transient, in parallel with animal pathologic recovery, and microbiota replacement on day 14 post-infection (10, 15). Whether intestinal microbiota recovery is

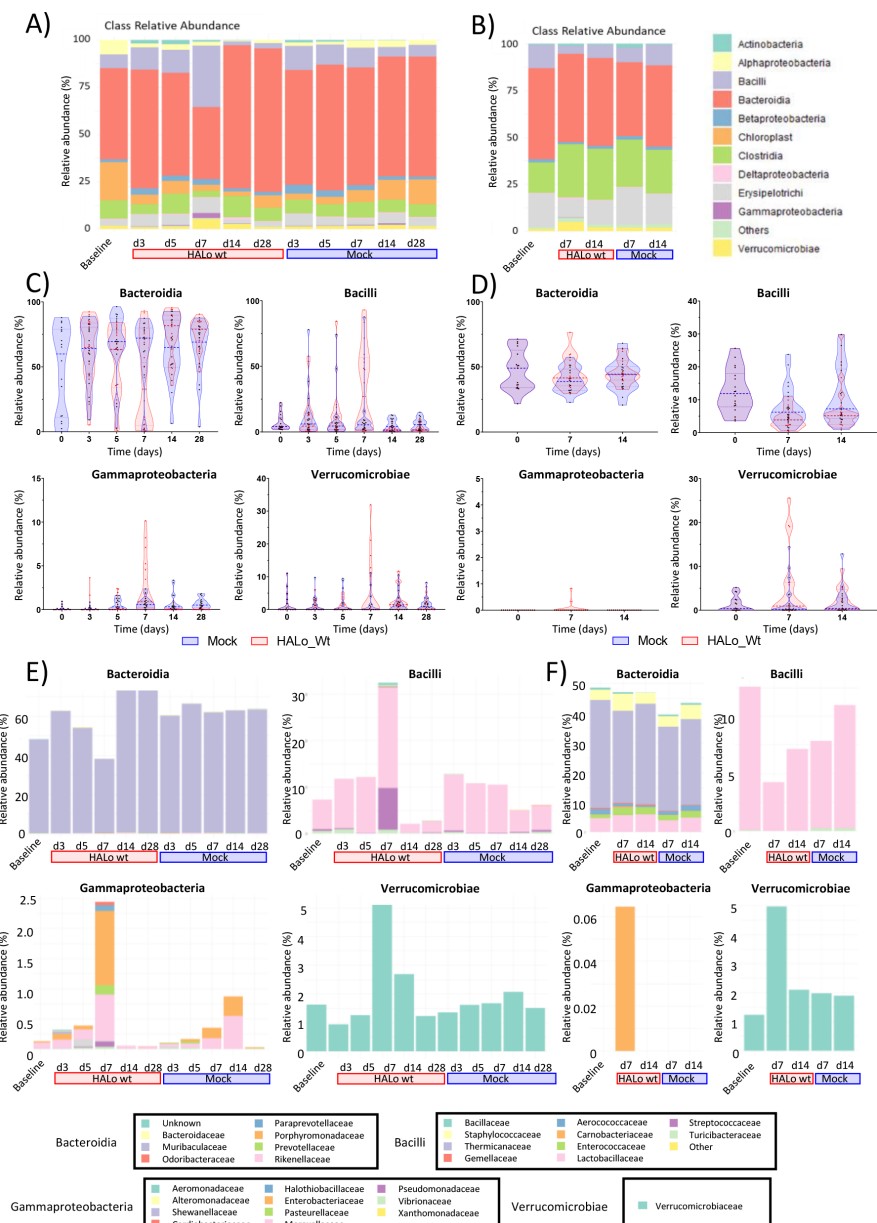

**FIG 4** Metagenomic analysis of small intestine and fecal microbiota. Bacteria Class relative abundance at different time points of (A) small intestine and (B) fecal samples from IAV-infected animals vs mock-infected animals previously described by Yildiz et al. Relative abundance (%) of the indicated bacterial class is depicted for mock (blue violin with blue dots) or IAV-infected (red violin with red dots) mice at different time points for (C) small intestine or (D) fecal samples. Orden relative abundance (%) of indicated bacterial class a different time points for (E) small intestine or (F) fecal samples. The legend to interpret colors associated with bacteria is shown in (E) and (F).

influenced by previously altered antibiotic-resistant phenotypes, as described in other contexts (35, 36) requires future characterization in the case of IAV infection. These changes could explain the differences in growth observed using cecal samples in Fig. 1. An additional caveat in the present study may be attributed to bacterial viability since the samples were subjected to freeze-thaw, thus influencing the metabolic profile and antibiotic resistance results. However, it can be fairly assumed that the community properties would be similarly affected in samples taken from mock-treated or IAV-infected animals. This study aims to specifically underline the community-level evaluation of microbiota rather than individual species properties.

The cenoantibiogram antibiotic resistance profile analysis presented here shows time-dependent differences in IAV-infected mice, specific to some but not all the antibiotics under analysis. This fact is an argument for using our experimental approach as an initial approximation to further characterize functional differences that could have significant implications. It is safe to assume that differences observed in metabolic state and antibiotic resistance could not be attributed to an inoculation bias since such an effect would cause global changes. Instead, specific changes occurring in a small subset of bacteria, or their expression pattern can explain such differences. In addition, changes in community properties presented here are in accordance with changes in the composition of certain bacteria previously reported by Yildiz et al. (15), whose cecal samples are analyzed in the present study.

Different studies have linked the increment of Gamma-proteobacteria, especially *Enterobacteriaceae,* to starvation or viral-induced dysbiosis with an increase in the expression of antibiotic resistance genes. *Enterobacteriaceae* could host some of these genes (12, 37, 38). Increased amounts of *Enterobacteriaceae* in the small intestine and feces (Fig. 4) suggest this possibility. In any case, the mere presence of certain bacteria could not be enough to express antibiotic resistance mechanisms that may require stimuli coming from the infection context.

Although there is an observation of a possible trend, due to the community-wide analysis performed here, this study could not present sufficient evidence for a direct causal link between nutrient uptake profiles and relative antibiotic resistance profiles. Future nutrient usage signatures, metabolomic analysis, and cenoantibiogram valida-tions from purposeful biased cecal samples would improve the validation of such links. Yet, the quality and robustness of the assays allow comparison of steady-state or altered bacterial communities harboring distinctive resistance elements, including the contribution to the phenotype by anaerobic and capnophilic bacteria.

Cecal samples from IAV-infected animals presented elevated changes in antibiotic resistance against some second-generation cephalosporins like cefoxitin, as well as those of the third generation (cefotaxime and ceftazidime) on days 3 to 5 and even for fourth generation's cefepime (Fig. 3C). Cephalosporin resistance increase can also be linked to an increased expression of β-lactamase phenotypes in *Enterobacteriaceae* (39, 40). Again, the use of metagenomic analysis will not be enough to determine the expression of these enzymes. In this scenario, possible inter-species resistance transmission (especially those transmissible through the exchange of mobile gene elements) cannot be ruled out (35, 41). In addition, the selection of antibiotic-resistant bacteria could be boosted clinically by concomitant antibiotic treatment.

Analyzing antibiotic resistance in the microbial cecal community is a challenge beyond the traditional concept of individual antibiotic resistance against one specific strain. The origin of antibiotic resistance at the microbial community level is difficult to discern and may not necessarily be linked to classical antibiotic resistance by specific bacteria populations. Instead, resistance may be linked to synergistic func-tional interactions between different bacteria species in a community, minimizing antibiotic activities, for example, by polymicrobial biofilms. If this is the case, interac-tions between non-pathogenic commensals could build changes in community-based antibiotic resistance that can be detected by a cenoantibiogram analysis.

Influenza virus as well as other relevant viruses producing acute infections such as SARS-CoV-2 could also affect intestinal microbiota functions that may determine systemic effects. Antibiotic resistance alterations such as the ones observed here open a new factor to be considered in the field of virus-host interactions specifically in the case of infections by influenza viruses. Some of the secondary infections associated with influenza are produced by bacteria that require oral antibiotic treatments, and how intestinal microbiota interacts with them may require further attention.

## MATERIALS AND METHODS

### Sample collection

Sampling was carried out by Yildiz et al. in 2018 (15) with C57BL/6J mice (8- to 9-week-old females). All animals were exposed to the same SPF / BSL2 (Specific Pathogen Free/ Biosafety Level 2 conditions) for 7 days feeding *ad libitum*. They were then treated as (i) Baseline or untreated animals, (ii) Mock or animals that were administered 40 µL of intranasal phosphate-buffered saline (PBS) without virus (control), and (iii) IAV or mice that were inoculated with virus A/VN/1203/2004 (low pathogenic variant) in 40 µL intranasal PBS solution. Upon reaching the experimental endpoints, the animals were humanely sacrificed using controlled exposure to $CO_2$. Organs were collected using sterile tools. Utensils were changed between each organ and each experimental group to avoid cross-contamination. Cecum samples extracted in an aerobiotic environment were stored at −70℃. All animal procedures were in accordance with federal regulations of the Bundesamt für Lebensmittelsicherheit und Veterenärwesen (BLV) Switzerland (Tierschutzgesetz) and approved by cantonal authorities (license number GE/44/17 and GE/45/15).

The experiments were performed using a very limited number of samples and cecal content from the animals used by Yildiz et al. (15). The reason for using these samples was based on the analysis of the sample complexity in a relevant animal model. In this paper, the authors performed a description of changes in the intestinal microbiota in the context of mice infected with recombinant attenuated Influenza H5N1 HALo (42). Using this model allows for studying influenza-associated changes in a non-lethal model, like the one that most influenza virus infections cause in humans.

### Sample processing

A mass of 0.4 g of intestinal content, free of tissue, was weighed for each sample of 40 mL. The feces were suspended in Falcon tubes with sterile 0.45% NaCl saline. They were heavily vortexed for 5 minutes and centrifuged for 5 minutes at 4,000 rpm. The pellet was discarded, and the supernatant was kept for successive tests.

### Semi-quantitative counting

A laminoculture (Uritest) was used for the standardization of the bacteria sample inoculum. For this, the suspension samples described above (0.4 g/40 mL) were seeded by immersion and incubated at 37℃ for 24 hours. After this time, the counting was carried out. For the assessment of total aerobic mesophiles, CLED medium was used.

### Community antibiotic resistance phenotypic profile (cenoantibiogram)

Given that there is no previous reference in the literature to the use of the cenoantibiogram on stool samples from experimental animals, it was necessary to adapt techniques traditionally used for susceptibility testing of microbial populations. We employed the Vitek 2 technique following the criteria of Robas et al. (17). One milliliter of bacteria suspension was diluted in 2 mL of sterile 0.45% saline solution. The 3 mL was used for the automatic inoculation of Vitek 2 AST-N243. Samples were incubated and automatically read by a Vitek 2 Compact, Automated ID/AST Instrument (Biomérieux, France).

### Metabolic profile of bacterial communities (CLPP)

The metabolic profile study was performed using Biolog EcoPlates. The plates used (BiologECO Biolog, Inc., Hayward, CA, USA) consist of 96 wells containing different sources of freeze-dried carbon and nitrogen, as well as a red-ox indicator of the consumption of said source (tetrazolium salts). The wells of the Biolog EcoPlates were loaded with 135 µL of the bacterial suspension prepared before from cecal material. The plates were incubated at 37℃ and absorbance was measured at 590 nm. Plate readings presented in the manuscript were made 72 hours after inoculation.

## DNA extraction, Library construction and Illumina sequencing, and Bioinformatics analysis

The methodology used for analyzing the small intestine and fecal microbiota was described by Yildiz et al. (15). Data processing, graphs, and statistical analysis were performed using the R 3.6.0 software.

16S rRNA gene NGS data were deposited at NCBI Bioproject (https://www.ncbi.nlm.nih.gov/bioproject/) under the accession numbers PRJNA419861 for SI samples, PRJNA419862 for fecal pellet samples, included already in the Materials and Methods section of Yildiz et al. (15).

## Data processing

The SPSS program (version 26.0 IBM Corp.) was used to perform the statistical analysis. Changes in the apparent number of bacteria growing in the laminoculture were performed using a two-way ANOVA followed by Sidak's multiple comparison test.

In the case of the metabolic profile, an initial multivariate analysis was performed by random forest (43) followed by Boruta algorithm analysis for featuring relevance estimation on top of an iterated (random forest) analysis. The results are represented in a box-whisker diagram. The algorithm is designed as a wrapper method. The original data set is extended by adding the so-called shadow features whose values are randomly permuted among the original cases to remove their correlations with a decision variable. This procedure makes it possible to establish a threshold defined as the highest feature importance recorded among the shadow features and a confidence interval regarding whether a factor is relevant. In addition, Boruta analysis was chosen to be able to capture non-linear relationships in a large number of substrates. Multiple comparisons between different groups were performed using a Wilcoxon-Mann-Whitney test to detect significant differences among them when $P < 0.05$.

The statistical analysis of the cenoantibiogram values was also performed by random forest followed by Boruta analysis to be able to capture non-linear relationships significantly different across samples. In addition, group condition comparisons were analyzed using a Wilcoxon-Mann-Whitney test to detect significant differences among them. A $P$-value $< 0.05$ was set as the limit to consider significant differences between conditions.

## ACKNOWLEDGMENTS

This work was supported by the Ministerio de Ciencia y Tecnología grant project PID2019-105761RB-100 to E.N.-V. This project was also funded by Swiss National Foundation grant project SNF310030_155949 to M.S. The authors acknowledge a FPI fellowship by Universidad San Pablo CEU to S.R.-R. J.A.-H. was supported by a PFIS fellowship co-funded by the FEDER/FSE and the ISCIII.

We would like to thank Sara Izpura-Luis and Brian Crilly for their editorial help.

## AUTHOR AFFILIATIONS

[1]Department of Pharmaceutical and Health Sciences School of Pharmacy, Microbiology Section, Departamento de Ciencias Farmacéuticas y de la Salud, Facultad de Farmacia, Universidad San Pablo-CEU, Madrid, Spain

[2]Facultad de Medicina, Instituto de Medicina Molecular Aplicada (IMMA), Universidad San Pablo-CEU, Madrid, Spain

[3]Department of Microbiology and Molecular Medicine, Medical Faculty, University of Geneva, Geneva, Switzerland

[4]Department of Microbiology, Icahn School of Medicine at Mount Sinai, New York, New York, USA

[5]CEMBIO (Centre for Metabolomics and Bioanalysis), Facultad de Farmacia, Universidad San Pablo-CEU, Madrid, Spain

[6]Geneva Center of Inflammation Research, Medical Faculty, University of Geneva, Geneva, Switzerland

## AUTHOR ORCIDs

Pedro A. Jiménez ⓘ http://orcid.org/0000-0003-0305-6889
Estanislao Nistal-Villan ⓘ http://orcid.org/0000-0003-2458-8833

## FUNDING

| Funder | Grant(s) | Author(s) |
| --- | --- | --- |
| Ministerio de Ciencia e Innovación (MCIN) | project: PID2019-105761RB-100 | Estanislao Nistal-Villan |
| Swiss National Science Foundation (SNF) | project: SNF310030_155949 | Mirco Schmolke |

## AUTHOR CONTRIBUTIONS

Marina Robas, Conceptualization, Data curation, Formal analysis, Investigation, Methodology, Resources, Validation, Writing – original draft | Jesús Presa, Data curation, Investigation, Software, Validation, Visualization, Writing – original draft | Javier Arranz-Herrero, Formal analysis, Software | Soner Yildiz, Formal analysis, Investigation, Methodology, Writing – original draft | Sergio Rius-Rocabert, Investigation, Writing – original draft | Francisco Llinares-Pinel, Formal analysis, Writing – original draft | Agustin Probanza, Conceptualization, Methodology, Supervision, Writing – review and editing | Mirco Schmolke, Conceptualization, Formal analysis, Investigation, Methodology, Supervision, Writing – original draft | Pedro A. Jiménez, Conceptualization, Data curation, Formal analysis, Funding acquisition, Methodology, Resources, Supervision, Writing – original draft | Estanislao Nistal-Villan, Conceptualization, Data curation, Formal analysis, Funding acquisition, Investigation, Methodology, Project administration, Resources, Supervision, Writing – original draft, Writing – review and editing

## ADDITIONAL FILES

The following material is available online.

Open Peer Review

**PEER REVIEW HISTORY (review-history.pdf).** An accounting of the reviewer comments and feedback.

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
