## [Reviewer comments · Microbiology Spectrum]

Microbiology Spectrum

Influenza A virus infection alters resistance profile of gut microbiota to clinically relevant antibiotics.

Estanislao Nistal-Villan, Marina Robas, Pedro Jiménez, Jesús-Luís Presa Matilla, Javier Arranz-Herrero, Mirco Schmolke, Sergio Rius Rocabert, Soner Yildiz, Francisco Llinares-Pinel, and Agustin Probanza

Corresponding Author(s): Estanislao Nistal-Villan, Universidad CEU San Pablo

Review Timeline:

Submission Date:	September 28, 2022
Editorial Decision:	February 23, 2023
Revision Received:	March 31, 2023
Editorial Decision:	July 4, 2023
Revision Received:	September 20, 2023
Editorial Decision:	October 7, 2023
Revision Received:	October 17, 2023
Accepted:	October 18, 2023

Editor: Alison Sinclair

Reviewer(s): Disclosure of reviewer identity is with reference to reviewer comments included in decision letter(s). The following individuals involved in review of your submission have agreed to reveal their identity: Karishma Bisht (Reviewer #2)

Transaction Report:

DOI: <https://doi.org/10.1128/spectrum.03635-22>

Dr. Estanislao Nistal-Villan
Universidad CEU San Pablo
Dpto. CC. Farmacéuticas y de la Salud
Madrid
Spain

Re: Spectrum03635-22 (Influenza A virus infection alters resistance profile of gut microbiota to clinically relevant antibiotics.)

Dear Dr. Estanislao Nistal-Villan:

I have received the review of your manuscript entitled "Influenza A virus infection alters resistance profile of gut microbiota to clinically relevant antibiotics.", and I regret to inform you that we will not be able to publish it in Spectrum.

Your submission was read by a reviewer with expertise in the area addressed in your study and it was the reviewer's view that your paper did not meet the standards necessary for publication. Specifically, some of the previous reviewers' major comments could not be addressed due to sample availability. We understand this but it limits the extent of the interpretation of the results within this study.

I am sorry to convey a negative decision on this occasion, but I hope that the enclosed reviews are useful. Please note, rejections from Microbiology Spectrum are final and your manuscript will not be considered by other ASM journals. We wish you well in publishing this report in another journal and hope that you will consider Spectrum in the future.

Sincerely,

Alison Sinclair
Editor, Microbiology Spectrum

Reviewer comments:

Reviewer #1 (Comments for the Author):

In the manuscript titled "Influenza A virus infection alters resistance profile of gut microbiota to clinically relevant antibiotics", the authors analyze the cecum's metabolic and antibiotic resistance profile from samples previously collected and analyzed in Yildez et al. The authors report a decreased bacterial load, changes in carbon and nitrogen sources, and changes in the antibiotic resistance of cecum microbes, particularly cefoxitin and ceftazidime. The study addresses the importance of understanding the underlying changes to nutrient preferences and the prevalence of antibiotic-resistant genes in the host gut microbiome during IAV infection.

Major Comments

1. While the authors discuss potential bacteria taxa, such as Enterobacteriaceae, to be linked to antibiotic resistance genes, the presence of this bacteria within this sample set is unknown. The sample type analyzed by Yildez et al. was the small intestine, which has been previously demonstrated to have a different microbial composition from the cecum. It would be ideal to report the microbial composition of the samples analyzed within these analyses or further discuss the differences in sample type between the two studies and its effect on the interpretation of the data.

2. The analyses performed within this study were on aerobic bacteria. However, the majority of the gut microbiome is composed of obligate anaerobic bacteria. The limitation of only analyzing aerobic bacteria should be further discussed as these results do not represent numerous bacteria within the intestinal microbiota.

3. Within the discussion, it is stated that the relative number of bacteria in laminoculture and bacteria load based on OTU abundance was consistent in order to counter the limitation of only analyzing aerobic bacteria. Is the OTU abundance referring to the samples analyzed in Yildiz et al? If so, Yildiz et al. analyzes the small intestine, which has been shown to have a different number of OTUs, diversity, and composition compared to the cecum, which is the sample type analyzed within this study.

Therefore, the two sample types are not an ideal comparison. It would be ideal to report the number of OTUs from the cecum samples used in this study.

Minor Comments

Within the introduction and the discussion, numerous paragraphs were 1-3 sentences in length. Suggest combining paragraphs so that the text is not disjointed.

Line 61. Suggest inserting (IAV) after "Influenza".

Line 125 -127: Suggest rewording.

Line 145-150: Suggest rewording and breaking into 2 sentences.

Line 200-204: Suggest rewording and breaking into multiple sentences.

Line 223-226: Suggest rewording.

Line 228-229: Suggest rewording.

Line 290-292: Suggest rephrasing. The references stated discuss the prevalence of antibiotic-resistant genes in a specific bacterial species in children with and without antibiotic treatment and adults. However, it is not suggested within the reference that the recovery of the microbiome from respiratory illness is influenced by antibiotic resistance phenotypes.

Table 1 Legend: Suggest stating the meaning of acronyms used within the table, such as BLEE.

July 4, 2023

Dr. Estanislao Nistal-Villan
Universidad CEU San Pablo
Dpto. CC. Farmacéuticas y de la Salud
Madrid
Spain

Re: Spectrum03635-22R1-A (Influenza A virus infection alters resistance profile of gut microbiota to clinically relevant antibiotics.)

Dear Dr. Estanislao Nistal-Villan:

Thank you for submitting your manuscript to Microbiology Spectrum. Your work has been evaluated by two experts in the field. Both reviewers were interested in your work and found it has merit. Reviewers also pointed out important aspects that must be addressed before the manuscript is ready for publication. Please, see below my recommendations below, together with the reviewers's report.

1) Two limitations are that the study does not provide any information about the taxonomy of the bacteria investigated and focuses only on aerobic microorganisms. These shortcomings raise the issue of whether the study offers a clear picture of the effect of IAV on the gut microbiome. As the requirement for publication in Microbiology Spectrum is technically sound work regardless of potential impact, your work could still be considered if you address the limitations of your study in the revised version.

2) Ensure the number of samples and the statistical analysis used is summarized in the figure legends and clearly explained in the figure legends. When overiewing reviewer 2 comments, I could not find the information requested.

Link Not Available

Sincerely,

Silvia Cardona

Journals Department
American Society for Microbiology
1752 N St., NW

Reviewer comments:

Reviewer #1 (Comments for the Author):

In the manuscript titled "Influenza A virus infection alters resistance profile of gut microbiota to clinically relevant antibiotics", the authors analyze cecum samples previously collected by Yildez et al. to analyze the metabolic and antibiotic resistance profile of culturable aerobic bacteria. The authors report a decreased bacterial load during peak viral titers within the lungs at 3, 5, and 7 days post-infection, and changes in the antibiotic resistance of cecum microbes, particularly ceftazidime, and ceftazidime. The study focuses on the importance of evaluating underlying changes to nutrient preferences and the prevalence of antibiotic-resistant genes during IAV infection.

Major Comments

While the authors provide interesting insight into the changes in nutrient preferences and antibiotic-resistant genes during IAV infection, there is a limited association with influenza virus infection. Since the major difference between the two groups being compared is influenza A virus infection, it would be informative to correlate the major findings reported in this paper with the virus shedding and pathogenicity observed post influenza A virus infection reported in Yildez et al. The majority of the significant findings within this paper occur between 3 and 7 dpi, which correlates with high viral loads in the lungs and particularly, weight loss and a large number of infiltrating immune cells within the lungs at 7 dpi.

Minor Comments

Line 74. Suggest removing "The" before influenza viruses

Line 84-88: Suggest combining this paragraph with the previous paragraph about influenza viruses in lines 74-78

Line 129-131: In addition to correlating with reduced genome counts in the small intestine, days 3 and 7 had high viral titers within the lungs of infected mice in Yildez et al.

Line 151-153: Similar to the comment above, since the major difference between the two groups is IAV infection, days 3 and 7 had the highest viral titers during infection which may be important to mention since the largest changes are observed at those days.

Line 157: Suggest adding "the" before substrates

Line 215 - 225: Suggest combining this paragraph with the previous paragraph

Reviewer #2 (Comments for the Author):

The manuscript by Robas et al focuses on the differences in the cecum's metabolic and antibiotic resistance profile from mice cecal samples that were exposed to mock or IAV infection. The authors have reported changes in viable bacterial load at different days post-infection. Additionally, they report changes in carbon and nitrogen sources as well as alterations in antibiotic resistance of cecal bacterial communities to specific antibiotics like cephalosporins. This is a relevant and interesting study where the authors have used microbiological assays to understand the effect of IAV infection on changes to nutrient preferences and the prevalence of antibiotic-resistant genes in gut microbiome. Overall, the manuscript is well-written and easy to follow. However, I have a few general comments for the authors:

Major Comments

1. The authors have neither analyzed nor have reported the microbial composition of the samples used in this study which makes it difficult to interpret the results. Especially since there is a lot of speculation made in the discussion section, it will be immensely helpful to pinpoint which bacterial species is in fact responsible for differences observed for the antibiotic resistance profile in mock versus IAV infected samples.
2. Figure 1: No error bars have been shown. Also, it will be useful to mention the number of samples that have been used to plot the bar graph.
3. Since the authors have only focused on aerobic bacteria in their study, which further raises the question if what they are observing is in fact giving a clear picture of the effect of IAV on the gut microbiome since a vast number of gut microbiome is composed of obligate anaerobic bacteria.

4. Figure 3B: Can the authors please justify the reason for plotting data for samples with a non-significant p-value
Figure 3C: please re-plot this data.

Overall, it will be helpful to see data points for bar graphs throughout the paper, as has been shown for Figure 3A.

Staff Comments:

Preparing Revision Guidelines

Please return the manuscript within 60 days; if you cannot complete the modification within this time period, please contact me. If you do not wish to modify the manuscript and prefer to submit it to another journal, please notify me of your decision immediately so that the manuscript may be formally withdrawn from consideration by Microbiology Spectrum.

Madrid, 28 of August, 2023

Dear Dr Cardona,

Below you can find our point-by-point response to your comments as well as the response to the reviewer's critiques. We have done our best to address all the issues you have raised.

Editor's comments:

Thank you for submitting your manuscript to Microbiology Spectrum. Your work has been evaluated by two experts in the field. Both reviewers were interested in your work and found it has merit. Reviewers also pointed out important aspects that must be addressed before the manuscript is ready for publication. Please, see below my recommendations below, together with the reviewers's report.

- 1) Two limitations are that the study does not provide any information about the taxonomy of the bacteria investigated and focuses only on aerobic microorganisms. These shortcomings raise the issue of whether the study offers a clear picture of the effect of IAV on the gut microbiome. As the requirement for publication in Microbiology Spectrum is technically sound work regardless of potential impact, your work could still be considered if you address the limitations of your study in the revised version.*

Thank you for the suggestions. In this current version, we present the metagenomic/taxonomic analysis of the small intestine material on days 3, 5, 7, 14, and 28 as well as the fecal material on days 7 and 14. We have included this analysis as Supplementary Figure 1 material. Although the bacteria found in this analysis are not from the cecum, we can infer that changes in bacteria composition occur across the digestive microbiota and that bacteria composition changes at the community level are connected to the functional changes described in the manuscript. Unfortunately, we could not analyze the bacteria composition and taxonomic analysis of the cecum samples used for the functional assays presented due to the sample availability.

When comparing the composition of the microbiota between Mock and HALO IAV infected mice, we can appreciate differences at day 7 in the small intestine (SI) that specifically affect Bacteroidia, Bacilli, Gammaproteobacteria, and Verrucomicrobiae classes (Panels A) and C) in Supplementary Figure 1). A more specific analysis (panel E) shows the prevalence of the Muribaculaceae family within the Bacteroidales order and Bacteroidia class. This family of bacteria is reduced at 7 dpi (days post-infection). Lactobacillaceae and Spreptococcaceae within the Lactobacillales order and Bacilli class are increased at 7 dpi. At the same time, the Gammaproteobacteria class shows an increase at 7 dpi. In particular Cardiobacteriales order and specifically the Cardiobacteriaceae family; Chromatiales order and specifically the Halothiobacillaceae family; the highest increase, was observed in the Enterobacteriales order and Enterobacteriaceae family; Pasteurellales order and specifically Pasteurellaceae family; high increase of Pseudomonadales order and specifically Moraxellaceae family; Vibrionales order and specifically Vibrionaceae family increases also. In the case of the Verrucomicrobiae, Verrucomicrobiales order, and Verrucomicrobiaceae family experience an increase also at 7 dpi. In panels C) and D), we can see data corresponding to individual mice (dots) that present considerable variability between animals.

Although upstream from the cecum, these differences in the small intestine between mice could explain the high variability of our functional assays.

Based on SI observations, we also analyzed the fecal microbiota, downstream from the cecum, on days 7 and 14 trying to see some correlation between fecal and the upstream microbiota at the SI. No clear differences could be observed in Bacteroidia, but a reduction in Lactobacillus at 7 dpi could be observed. A small but significant increase in the Gammaproteobacteria and Verrucomicrobiae classes can be observed at 7 dpi (Panel F Supp Fig 1). This correlation between SI and fecal microbiota could indicate a similar phenotype at the cecum, used for the functional assays. As mentioned in the manuscript Discussion section, Enterobacteriaceae belonging to Gammaproteobacteria (lines 313, 380, and 397) have been linked to overgrowth and increased antibiotic resistance. However, 16S gene analysis cannot determine whether this is the source of the resistance. Metagenomic analyses, cannot guarantee the identification of the members in the communities responsible for the functional changes analyzed in the manuscript either. The presence or absence of specific taxa can be related to particular functions, but the experimental determination of such functions in native conditions within the community complexity is a challenge to be addressed in the future.

We have included in different places across the manuscript the limitation of extracting, manipulating, and culturing the microorganisms in the samples under aerobic conditions. We understand the limitations of this approach however, all the samples were manipulated the same way and the changes observed are indicative of metabolic and antibiotic resistance changes that could be further described in the future by considering also anaerobiosis. This limitation does not subtract from the relevance of functional changes at the community level and could be measured and considered to interpret phenotypes. Reproduction of the native environment of the microorganisms present in the cecal samples is not trivial and will require culture conditions to be defined, particularly the culture media and the atmospheric conditions.

- 2) *Ensure the number of samples and the statistical analysis used are summarized in the figure legends and clearly explained in the figure legends. When overviewing reviewer 2 comments, I could not find the information requested.*

We appreciate this observation. We have made a thorough rewriting of figure legends trying to clarify the analysis conditions, including the number of samples in each experiment presented in the figures.

We'd also like to highlight that, in response to a suggestion from Reviewer 2, we have incorporated individual values into the analysis. During this process, we identified an error in the duplication of nutrient uptake data from day 3 in samples obtained from IAV-infected animals. After rectifying this duplication, we proceeded to update the panels in Figure 2 and adjusted the accompanying result descriptions accordingly.

Reviewer comments:

Reviewer #1 (Comments for the Author):

In the manuscript titled "Influenza A virus infection alters resistance profile of gut microbiota to clinically relevant antibiotics", the authors analyze cecum samples previously collected by

Yildez et al. to analyze the metabolic and antibiotic resistance profile of culturable aerobic bacteria. The authors report a decreased bacterial load during peak viral titers within the lungs at 3, 5, and 7 days post-infection, and changes in the antibiotic resistance of cecum microbes, particularly cefoxitin, and ceftazidime. The study focuses on the importance of evaluating underlying changes to nutrient preferences and the prevalence of antibiotic-resistant genes during IAV infection.

Major Comments

- *While the authors provide interesting insight into the changes in nutrient preferences and antibiotic-resistant genes during IAV infection, there is a limited association with Influenza virus infection. Since the major difference between the two groups being compared is Influenza A virus infection, it would be informative to correlate the major findings reported in this paper with the virus shedding and pathogenicity observed post-influenza A virus infection reported in Yildez et al. Most of the significant findings within this paper occur between 3 and 7 dpi, which correlates with high viral loads in the lungs and particularly, weight loss and a large number of infiltrating immune cells within the lungs at 7 dpi.*

We appreciate the reviewer's comment. We have included this point addressing some of the minor comments below as well as the discussion, where both viral titers, as well as pathogenicity described by Yildiz et al., are correlated with the observations made in our manuscript.

Minor Comments

- *Line 74. Suggest removing "The" before influenza viruses.*
Thank you for the suggestion. The text has been fixed.
- *Line 84-88: Suggest combining this paragraph with the previous paragraph about influenza viruses in lines 74-78*
Thanks for suggesting. Paragraphs have been merged.
- *Line 129-131: In addition to correlating with reduced genome counts in the small intestine, days 3 and 7 had high viral titers within the lungs of infected mice in Yildez et al.*
Thank you for the suggestion. We have addressed this suggestion in the manuscript.
- *Line 151-153: Similar to the comment above, since the major difference between the two groups is IAV infection, days 3 and 7 had the highest viral titers during infection which may be important to mention since the largest changes are observed at those days.*
Following the reviewer's suggestion, a comment about the high viral titers on days 3 and 7 has been included in the indicated location of the manuscript.
- *Line 157: Suggest adding "the" before substrates.*
Thank you for the suggestion, the text has been changed.
- *Line 215 - 225: Suggest combining this paragraph with the previous paragraph.*

Thanks for noticing. The idea of splitting paragraphs was to emphasize changes in different antibiotic families, however, following the reviewer's suggestions, the paragraphs between lines 215 and 225 have been merged with the previous paragraph.

Reviewer #2 (Comments for the Author):

The manuscript by Robas et al focuses on the differences in the cecum's metabolic and antibiotic resistance profile from mice cecal samples that were exposed to mock or IAV infection. The authors have reported changes in viable bacterial load at different days post-infection. Additionally, they report changes in carbon and nitrogen sources as well as alterations in antibiotic resistance of cecal bacterial communities to specific antibiotics like cephalosporins. This is a relevant and interesting study where the authors have used microbiological assays to understand the effect of IAV infection on changes to nutrient preferences and the prevalence of antibiotic-resistant genes in gut microbiomes. Overall, the manuscript is well-written and easy to follow. However, I have a few general comments for the authors:

We appreciate the reviewer's comment about the value of the work and the effort to present the analysis in a way easy to follow.

Major Comments

- 1. The authors have neither analyzed nor reported the microbial composition of the samples used in this study which makes it difficult to interpret the results. Especially since there is a lot of speculation made in the discussion section, it will be immensely helpful to pinpoint which bacterial species is, in fact, responsible for differences observed for the antibiotic resistance profile in mock versus IAV-infected samples.*

Thank you for the comment. As pointed out above in response to the editor's comments, the bacteria composition and taxonomic analysis of the samples used in the functional assays could not be performed due to sample availability. Instead, we are presenting the metagenomic analysis of small intestine (upstream of cecum) composition at days 3, 5, 7, 15, and 28 of mock and IAV-infected mice. Although they may present differences compared to the bacteria composition at the cecum, we can observe changes in composition that correlate with the pathology observed in the animal. This data could be an indicator and a reference to establish the changes occurring during infection.

In addition, since the largest changes are occurring at day 7 dpi in the classes already mentioned, we could compare the composition of the fecal microbiota, just after the cecum at days 7 and 14 post-infection. The increased amount of Enterobacteriaceae at 7 dpi in IAV-in the small intestine and fecal samples, upstream and downstream of the place where the samples were obtained for the functional assays of our manuscript. As stated before, an Enterobacteria increase could be associated with an increase in beta-lactam resistance increase. However, this is just speculation, and more effort in developing functional assays is needed.

Despite the composition containing the bacteria responsible for the functional changes described in the manuscript, this type of analysis cannot identify the bacteria or group of bacteria responsible for the changes in nutrient uptake and cenoantibiogram resistance.

This association remains a challenge in future descriptions of microbial community functional changes, and we hope to address specific associations in the future.

2. *Figure 1: No error bars have been shown. Also, it will be useful to mention the number of samples that have been used to plot the bar graph.*

Thank you for noticing. When error bars are not presented, it means that the detection limit of the assay has been reached. No sample was available to repeat the analysis, however, the analysis allows us to detect differences in the amount of the bacteria able to grow in the given experimental conditions between IAV and mock-infected mice. The number of samples for each condition was 3. It has been included in the text.

3. *Since the authors have only focused on aerobic bacteria in their study, it further raises the question if what they are observing is giving a clear picture of the effect of IAV on the gut microbiome since a vast number of gut microbiome is composed of obligate anaerobic bacteria.*

We appreciate this comment. As indicated above in the editor's general comments, we have included in several places across the manuscript the caveat of manipulating samples in an aerobic atmosphere. This limitation was applied to all the samples that would suffer the same aerobic restrictions. Still, we can see specific nutrient uptake and antibiotic resistance changes between IAV and mock-infected animals, which is one main argument in this manuscript. We cannot be sure that these functional changes will be the ones occurring inside the animals, but functional changes are occurring at the community level when comparing IAV and mock-infected animals at certain days post-infection. Some of them may be biologically relevant. Not only anaerobic bacteria may suffer by working in aerobic conditions, but also culture media, the presence of other microorganisms in the community, or host factors from the mice can influence functional changes. Addressing these issues will require technical improvements in the future. The main message of our work focuses on the fact that IAV infection causes functional alterations in the cecum (and probably other intestinal parts) microbiota at the community level with potential biological relevance.

4. *Figure 3B: Can the authors please justify the reason for plotting data for samples with a non-significant p-value.*

Thank you for the comment. The reason to include these values is to have a reference for those that indeed present differences. The statistical analysis was systematically performed for each IAV vs. mock-infected group. All the comparisons for the conditions under study were analyzed and presented as shown in the panel.

5. *Figure 3C: Please re-plot this data. Overall, it will be helpful to see data points for bar graphs throughout the paper, as has been shown in Figure 3A.*

Thank you for your suggestions. Individual values have been added to the figures. We want also to note that this suggestion allowed us to identify a duplication of one mouse sample in the IAV-infected group. As mentioned before, we have corrected this mistake and present Figure 2 panels B) and C) with these updated changes also in the text.

October 7, 2023

Dr. Estanislao Nistal-Villan
Universidad CEU San Pablo
Dpto. CC. Farmacéuticas y de la Salud
Madrid
Spain

Re: Spectrum03635-22R2 (Influenza A virus infection alters resistance profile of gut microbiota to clinically relevant antibiotics.)

Dear Dr. Estanislao Nistal-Villan:

Thank you for submitting your manuscript to Microbiology Spectrum. As you will see your paper is very close to acceptance. Please modify with the minor changes the manuscript one reviewer recommended. As these revisions are quite minor, I expect that you should be able to turn in the revised paper in less than 30 days, if not sooner. I will ensure your revised manuscript is considered for acceptance as soon as possible. Also, ensure a data availability statement is provided with a link to the repository where the data is deposited.

When submitting the revised version of your paper, please provide (1) point-by-point responses to the issues raised by the reviewers as file type "Response to Reviewers," not in your cover letter, and (2) a PDF file that indicates the changes from the original submission (by highlighting or underlining the changes) as file type "Marked Up Manuscript - For Review Only". Please use this link to submit your revised manuscript. Detailed instructions on submitting your revised paper are below.

Link Not Available

Sincerely,

Silvia Cardona

Reviewer comments:

Reviewer #2 (Comments for the Author):

The authors have addressed all my prior comments. Three minor comments remain:

- 1) The authors have to upload the metagenomic data to NCBI.
- 2) Figure 1 Day 3 data: I am not sure if the stats do justice here, as per your data, n=1 for your mock, while n=3 for your infected samples. Please mention which statistical test was used for Figure 1.
- 3) Additionally your supplementary figure 1 could be part of the main manuscript since it does show important parameters, regarding species diversity as well as species abundance.

Preparing Revision Guidelines

Please return the manuscript within 60 days; if you cannot complete the modification within this time period, please contact me. If you do not wish to modify the manuscript and prefer to submit it to another journal, please notify me of your decision immediately so that the manuscript may be formally withdrawn from consideration by Microbiology Spectrum.

Madrid, 15 of October, 2023

Dear Dr Cardona,

Below you can find our point-by-point response to your comments as well as the response to the reviewer's critiques. We have done our best to address all the issues you have raised.

Editor's comments:

Reviewer #2 (Comments for the Author):

Three minor comments remaining:

1) The authors have to upload the metagenomic data to NCBI.

"16S rRNA gene NGS data were deposited at NCBI Bioproject (<https://www.ncbi.nlm.nih.gov/bioproject/>) under the accession numbers PRJNA419861 for SI samples, PRJNA419862 for fecal pellet samples"

This is in the M&M of the Microbiome paper Yildiz S et al 2018

2) Figure 1 Day 3 data: I am not sure if the stats do justice here, as per your data, n=1 for your mock, while n=3 for your infected samples. Please mention which statistical test was used for Figure 1.

We are sorry if it was not clear in the figure legends. All the data for each group comes from triplicates. We are switching the figure, including individual values to make it more clear to appreciate. Please, notice that these numbers are estimated approximations based on growth interpretations of the laminoculture.

3) Additionally your supplementary figure 1 could be part of the main manuscript since it does show important parameters, regarding species diversity as well as species abundance.

Thank you for the suggestion. We are changing the text in the manuscript to include this figure as Figure 4.

October 18, 2023

Dr. Estanislao Nistal-Villan
Universidad CEU San Pablo
Dpto. CC. Farmacéuticas y de la Salud
Madrid
Spain

Re: Spectrum03635-22R3 (Influenza A virus infection alters resistance profile of gut microbiota to clinically relevant antibiotics.)

Dear Dr. Estanislao Nistal-Villan:

Your manuscript has been accepted, and I am forwarding it to the ASM Journals Department for publication. You will be notified when your proofs are ready to be viewed.

Sincerely,

Silvia Cardona
Editor, Microbiology Spectrum
